# Current Trends in Diagnostics of Viral Infections of Unknown Etiology

**DOI:** 10.3390/v12020211

**Published:** 2020-02-14

**Authors:** Daniel Kiselev, Alina Matsvay, Ivan Abramov, Vladimir Dedkov, German Shipulin, Kamil Khafizov

**Affiliations:** 1FSBI “Center of Strategic Planning” of the Ministry of Health, 119435 Moscow, Russia; neurolynx13@gmail.com (D.K.); arity767@gmail.com (A.M.); abriv@bk.ru (I.A.); shipgerman@gmail.com (G.S.); 2I.M. Sechenov First Moscow State Medical University, 119146 Moscow, Russia; 3Moscow Institute of Physics and Technology, National Research University, 117303 Moscow, Russia; 4Pasteur Institute, Federal Service on Consumers’ Rights Protection and Human Well-Being Surveillance, 197101 Saint-Petersburg, Russia; vgdedkov@gmail.com; 5Martsinovsky Institute of Medical Parasitology, Tropical and Vector Borne Diseases, Sechenov First Moscow State Medical University, 119146 Moscow, Russia

**Keywords:** viruses, viral infections, diagnostics, sequencing, PCR, NGS, HTS, single-molecule sequencing, bioinformatics

## Abstract

Viruses are evolving at an alarming rate, spreading and inconspicuously adapting to cutting-edge therapies. Therefore, the search for rapid, informative and reliable diagnostic methods is becoming urgent as ever. Conventional clinical tests (PCR, serology, etc.) are being continually optimized, yet provide very limited data. Could high throughput sequencing (HTS) become the future gold standard in molecular diagnostics of viral infections? Compared to conventional clinical tests, HTS is universal and more precise at profiling pathogens. Nevertheless, it has not yet been widely accepted as a diagnostic tool, owing primarily to its high cost and the complexity of sample preparation and data analysis. Those obstacles must be tackled to integrate HTS into daily clinical practice. For this, three objectives are to be achieved: (1) designing and assessing universal protocols for library preparation, (2) assembling purpose-specific pipelines, and (3) building computational infrastructure to suit the needs and financial abilities of modern healthcare centers. Data harvested with HTS could not only augment diagnostics and help to choose the correct therapy, but also facilitate research in epidemiology, genetics and virology. This information, in turn, could significantly aid clinicians in battling viral infections.

## 1. Introduction

As some statistical models suggest, there are over 320,000 mammalian viruses in existence [1], a little over 200 of which are known to infect humans [2], with the number increasing steadily over decades [3]. The recent years have been marked by a rising frequency of emerging viral infections, e.g., SARS [4], Zika [5], or a more recent *Coronavirus* 2019-nCoV outbreak [6]. Bearing that in mind, the range of available diagnostic tools seems disproportionate to the growing number of diseases, because they detect only known pathogens, which constitute ~0.07% of viral entities. Metaphorically, we are attempting to look at a vast sea of threats through the eye of a needle.

As of this day, a multitude of clinical tests are available for detection of viruses, e.g., FISH, ELISA (enzyme-linked immunosorbent assay), and PCR- and their numerous modifications. The latter has become a gold standard, for example, in cases of gastrointestinal [7] and respiratory infections [8]. Indeed, the presence of viral nucleic acids in biological media, particularly in blood plasma, while not definitive, is highly suspect of an ongoing infection, because it strongly implies viral replication. Furthermore, PCR not only enables us to detect pathogens, but also quantify them with great precision, based on the number of copies of a pathogen’s DNA/RNA templates in a sample.

As effective as they are, PCR tests are sometimes prone to major pitfalls, chiefly owing to high mutability of a viral genome, which significantly perplexes primer design and requires researchers to update primer sequences continually. With the entirety of human-infecting viruses, most of which are yet to be revealed, it becomes increasingly difficult to maintain up-to-date primer panels while also ensuring the correct reaction conditions and high specificity for differentiating closely related viral species. This situation is aggravated by the low rates of correct pathogen identification [9].

Immunoassays have been extensively used in clinical laboratories for over 60 years now. Fast and relatively reliable screening tests for infectious diseases have been successfully introduced into healthcare practice, e.g., for HIV, HBV, HEV, HAV, HCV, and Rubella virus. Although ELISA does not quantify pathogens with great precision, it effectively detects pathogens based on their antigen structure. Useful as they are, immunoassays are also subject to biases and fundamental pitfalls, although they are often disregarded for the sake of low cost and convenience.

Problems with conventional diagnostic methods have been stated in several studies. For example, Arnold et al. [10] found that only 62% of viral respiratory infections among children could be confidently attributed to known pathogens, mostly to human metapneumovirus (63%) and adenoviruses (45%), while almost a third of all cases remained undiagnosed. In another study by Vu et al. [11], it was demonstrated that in 42.7% of cases of viral gastroenteritis the pathogen could not be identified with conventional methods. Kennedy et al. [12] outline that up to 40% of viral encephalitis infections remain undiagnosed with modern clinical tests.

High throughput sequencing (HTS) represents a range of technologies based on sequencing by synthesis, which allow retrieving of multiple nucleotide sequences at once with high reliability for further pathogen identification. Thus, HTS may tackle the aforementioned diagnostic issues, adding an extra layer of data for identification and further insight into a pathogen.

In this paper, we assess modern clinical tests for viral infections and explore the ways that HTS provides detailed data about a pathogen’s genome, allowing for a deeper analysis of its features, such as drug resistance and phylogeny. We also hope to demonstrate how modern clinical laboratories would benefit from adopting HTS as a routine diagnostic technique.

## 2. Traditional Methods of Diagnosing Infections

Before molecular and serological diagnostics became widely accessible, detection of pathogens primarily relied on conventional microbiological methods, e.g., cultures on growth media, in cells, and in laboratory animals with further microscopy, identification of antigenic and pathogenic characteristics and the metabolic profile.

ELISA has become one of the most widely used tests, owing to high sensitivity and specificity (both usually reported at >90% [13,14,15]), low price per sample, and the simplicity of sample processing and result interpretation. Modern commercial kits are designed to cover a large variety of antigens, bacterial and viral, even at femto- and attomole concentrations. All these factors contribute to the popularity of ELISA, turning it into a universal tool for disease screening and prognosis. At the same time, strong limitations restrict its use. Because the test basically relies on antigen detection, cross-reactivity between heterologous antigens can lead to false positive results, like with influenza virus-specific CD8+ T cell and hepatitis C virus antigens [16]. Other fundamental methodological flaws have been described by Hoofnagle and Wener (2009) [17] and include the hook effect, possible presence of anti-reagent and autoantibodies and lack of concordance. It is also of note that ELISA does not provide any extra data beyond approximate viral burden and limited detection of a pathogen, rendering the identification of new viruses impossible.

The development of molecular cloning and commercial solutions for Sanger sequencing aided the identification of new pathogens. The core principle of this approach includes three steps: (1) amplification of a selected region via PCR, (2) molecular cloning of the amplicons, and (3) Sanger sequencing of the molecular clones. In some cases, hybridization is used to detect viral nucleic acids. This requires at least some knowledge of the target sequences, as it utilizes a microarray with oligonucleotides nested on its surface that are complementary to the conservative sequences in the genomes of the chosen viral species. The technique never grew to become popular in diagnostics, because (1) content of viral nucleic acids in the sample had to be high, and (2) high levels of the host’s DNA and the DNA of the microbiome interfered with amplification [18].

A number of alternative approaches have been proposed to detect target sequences in the pathogen’s genome. Among them are SISPA (sequence-independent single primer amplification) and its modifications [19,20,21,22,23], and VIDISCA (virus discovery based on cDNA-AFLP), which is based on the AFLP (amplified fragment length polymorphism) method that was specifically adapted for studying viromes [23,24].

In spite of a multitude of available technologies, real-time PCR remains, as of now, one of the most powerful and common, often regarded as a reference diagnostic test. RT-PCR has been extensively used [25] for studying viral genera [1,25,26,27], for which a significant number of primers have been designed [28,29,30]. Since the potential of this method became apparent, many enterprises have stepped in, developing commercial PCR kits for reliable fast-track diagnosis of infections like enterovirus, influenza, herpes simplex, etc. While effective and highly sensitive, these panels by design cannot identify new strains and types with altered target sequences. Because of possible genetic similarities between known and previously undiscovered viruses, primers might anneal incorrectly, resulting in false positives and taxonomic misidentification. Multiplexing of primers and probes can be challenging, as there is always a risk of cross-specificity and primer-dimer formation. While separating reactions solves this issue, it requires greater volumes of clinical samples, which are usually limited. Furthermore, in the case of probes, the number of targets is strictly limited to a handful of available fluorophores.

The choice of a diagnostic test is normally based on the circumstances. For instance, while popular and simple methods (PCR, ELISA) are quick, they are less informative than Sanger or next generation sequencing, and it is obvious that the latter could supply extra information about pathogens. (Figure 1)

## 3. Studying Viral Pathogens with High Throughput Sequencing (HTS)

The invention of HTS—alias next generation sequencing (NGS)—has revolutionized research in biology and medical science, including virology, allowing new viruses to be genetically analyzed without prior cultivation. Sequencing of viral genomes has already become a pivotal part of virological research [31], aiding clinicians in identifying complex infections [32]. It is gradually growing to become one of the main methods in molecular diagnostics [33], development of vaccines [34,35] and searches for new therapeutic strategies [36,37]. Over the years, it has pushed the frontiers of molecular epidemiology [38] and evolutionary genomics [39,40].

Another type of NGS-based analysis is metagenomics, which was comprehensively defined by Chen and Pachter as “the application of modern genomics technique without the need for isolation and lab cultivation of individual species” [41]. In essence, it offers the means to investigate the taxonomic diversity of a microbial community in an environmental or biological sample as the entirety of present nucleic acids is studied. Further applications of this method extend to transcriptomics and proteomics, allowing for a deeper insight into several aspects of pathogens, such as drug resistance, adaptation mechanisms and vast communication networks between species.

### 3.1. Metagenomic Approach

Metagenomics allows for identification and in-depth studies of unknown viral pathogens by using shotgun sequencing [42,43,44], which has been extensively applied in clinical and environmental research [45,46]. Nucleic acids from a sample undergo virtually unbiased sequencing, i.e., with minimum prejudice towards specific organisms [47,48]; in theory, the method can be used to analyze a potentially unlimited range of targets [49,50]. Nevertheless, there is evidence that metagenomic sequencing is restricted by multiple pitfalls and biases, based on the pathogen’s structure, extraction method, GC content, and other factors [51,52,53]. Therefore, there are five major limitations: (1) sequences of interest should share at least low identity with the analogous sequences in a reference genome, ensuring correct mapping; (2) the analytical complexity of obtained data often requires employment of a qualified bioinformatician and the use of a specialized computational infrastructure (including both hardware and software; (3) hidden experimental and methodological prejudices towards certain taxa; (4) defining the method’s sensitivity and properly measuring it against a relevant reference; (5) dealing with the abundance of host-cell, bacterial and fungal nucleic acids.

Unlike traditional tests [54], metagenomics removes the necessity of designing and synthesizing PCR primers and probes. This reduces time consumption, which is critical during outbreaks of viral infections, such as Zika virus [55] or Ebola virus [56], when fast unbiased pathogen identification is crucial for effective disease containment. Unbiased HTS can also aid in investigating cases of unknown clustered viral infections, when other diagnostic tools do not provide sufficient information, like in the case of the new *Arenavirus* [57].

A metagenomic approach is deemed most helpful when heterogeneous infections share an identical clinical presentation [58] or when genetic markers of antibiotic resistance are in question [50]. Data supplied using NGS also augment diagnostics of respiratory infections: whereas modern multiplex molecular assays reveal the etiology of respiratory infections in approximately 40–80% of cases [58,59,60,61,62], metagenomics successfully tackles this limitation [63].

Furthermore, some respiratory viruses (e.g., *Rhinoviruses*) frequently appear to be the only or the most abundant pathogens in samples from patients with respiratory infections [64]. This pattern implies that either the diagnostic tools are erroneous [65,66,67,68] or the groups of related viruses cause infections with a variable clinical manifestation [69]. Metagenomics reveals actual etiological factors, allowing for genotyping, identifying antibiotic resistance markers [70] and supplying data molecular epidemiology [71]. Recently, the applicability of HTS protocols to analyzing clinical samples has been evaluated, demonstrating high potential benefits for clinical studies [58,59,72].

Graf et al. (2016) [47] compared the sensitivity of two pathogen identification methods: metagenomic RNA sequencing (RNA-Seq) with further Taxonomer processing and commercial FDA-approved respiratory virus panel (RVP) for GenMark eSensor. They used 42 virus-positive controls from pediatric patients and 67 unidentified samples [47]. Metagenomic analysis revealed 86% of known respiratory pathogens, with supplementary PCR corroborating the finding in a mere third of discordant samples. However, for unknown samples, consistency between the two methods reached as high as 93%. Still, metagenomic analysis uncovered 12 extra viruses that were either not targeted by the RVP or failed to bind to complementary nucleotide strands on the chip because of significantly mutated genome sequences. A metagenomic approach not only aids in identification of a pathogen, but also grants an insight into its nucleotide sequence. Although viral sequences constituted only a small fraction of total reads, sufficient data for 84% of samples were gathered to allow for high-resolution genotyping.

Therefore, despite technological encumbrances, a metagenomic approach has a great potential for clinical application. It could provide the means to identify both known and novel pathogens, establish phylogenetic connections and detect drug-resistant and highly virulent strains of common viruses [47].

### 3.2. Problems of Metagenomic Approach

Even though the cost per raw megabase of DNA sequence has plummeted over the years [73], HTS remains a costly and time-consuming method, requiring a complex procedure of sample preparation, expensive equipment and at least some bioinformatic training with laboratory personnel, therefore rendering it unsuitable for large-scale screening studies [74]. For example, library preparation in the aforementioned study by Graf et al. took no less than 14 h. A HiSeq 2500 (Illumina, San Diego, California, USA) was used to conduct sequencing that lasted 11 consecutive days. Hypothetically, a metagenomic approach could be adapted to a clinical lab routine, provided that library preparation and run processing are optimized to deliver results as fast as conventional tests and present them in an easily interpretable form. Outsourcing of data processing is one option to solve this problem. Ranges of bioinformatics tools facilitate data analysis, e.g., Kraken [75], Kaiju [76], VirusFinder [77]. Different workflows have recently been benchmarked and comprehensively overviewed elsewhere [78,79,80].

In-house analysis is another alternative, given that the algorithms for sequence processing include criteria that would return a reliable interpretation of assessed reads. Ideally, such algorithms would not only filter out artefacts and perform mapping to the reference genome, but also suggest the most represented pathogen in each sample, narrowing down the spectrum of considered pathogens. This would create a fast definitive NGS-based diagnostic tool [81,82].

Library preparation requires the use of total DNA/RNA from a sample, implying the presence of nucleic acids from other organisms. Direct shotgun sequencing (i.e., without contaminant nucleic acid removal) yields most reads from a host, whereas target reads constitute a mere fraction of total data [50], leading to a significant drop in the method’s sensitivity. Researchers are forced to search for target sequences amongst a handful of “junk” reads, employing time and computer power to solve this “needle in a haystack” sort of challenge.

A few strategies can tackle this methodological nuisance. In principal, they are based on either one or both of the following approaches [83]: (1) in silico separation of host sequences based on their similarity with the hosts reference genome; (2) conducting of taxonomic classification for the whole set of sequences using comprehensive databases which includes sequences of both target and host genome. The former option is very demanding in terms of computational power, since millions of reads undergo a complex analysis [84]. The latter is subject to faults of its own: partial or low-quality reference sequences, false mapping and high genetic diversity resulting in numerous polymorphisms [79].

A pathogen’s content in a sample is unknown a priori and can only be approximated, for example, based on severity of an onset [85]. Consequently, it becomes difficult to evaluate a minimally required number of sequencing reads per sample (SRPS) sufficient for a reliable in-depth analysis. Underestimation of SRPS results in a low yield of target sequences, for example, 0.008% for Epstein-Barr virus (EBV) [86], 0.0003% for Lassa virus [87] and 0.3% for Zika virus [88]. Considering that metagenomics focuses on relative quantities of nucleic acids rather than absolute counts, establishing appropriate thresholds for positive and negative results becomes challenging. In cases of cell cultures and tissues, calculating copies per cell (CPC) has been suggested [89]; however, this parameter would not work for other types of samples, e.g., swabs and washes. High content of a host’s nucleic acids might artificially lower CPC, falsely attributing negative results to samples. As we have mentioned, the sequencing depth is another aspect to this issue, because shallow sequencing provides little data with poor scalability, to the point where an extra round of sequencing is required to verify the results based on low SRPS data [54]. Therefore, taking the aforementioned factors into account and designing a functional SRPS assessment tool poses an important objective for further research.

Furthermore, DNA/RNA samples might be subject to contamination, which creates artifacts [90]. Thus, it is crucial to keep track of any possible contamination within the laboratory. It would also help to scan scientific magazines periodically for the discovery of new microorganisms in normal human microbiota to adjust the sequence data accordingly.

Metagenomics was initially suggested for expanding the knowledge on existing families, rather than discovering new ones [91], which naturally limits its application to the broad-range search. In practice, reads very often fail to map to reference genomes due to a lack of even minimally homologous sequences [72]. This situation is anticipated in cases of organisms that have not been screened for viral infections before or whenever strains of known viruses in the sample possess unique sequences that have not been described yet. Thus, a better understanding of the virosphere requires not only expansive sets of samples, but also improvements to data processing algorithms. Paradoxically, the more we explore the biology of viruses, the more apparent it becomes that only a negligible part of their taxonomy is being studied and, to make matters worse, mainly that of closely related viruses, rather than the principally new ones [92,93]. In this way, while rapidly gaining profoundness, our knowledge of viruses is progressively narrowing.

Standardization of the method is yet another issue. Because metagenomics is aimed at describing a multitude of microorganisms in a sample, and also due to biases and host-cell DNA and RNA abundance, robust references and normalization methods have to be developed. A possible solution has been proposed by Wilder et al. [94], who suggested creating reference metagenomic libraries for sequencing with unknown samples. Indeed, this approach seems logical and reliable in cases when an abundance of a particular virus is expected. For example, comparing reference samples with a clinical sample, where an abundance of a particular viral species is observed, would indicate a possible on-going infection. Nevertheless, confidence intervals have to be set to ensure that the differences are significant, and studying more samples is required for an accurate evaluation, so further research is required. However, it is challenging to conceptualize standards when the search for new viruses is concerned, because there is no apparent candidate for a standard. In this case, limits on the sequencing quality can be imposed that would help differentiate between sequencing artifacts and the actual discovery of a new pathogen. Validation could include cultivation with supplementary PCR and further whole genome sequencing for new pathogens.

Finally, complex techniques of sample preparation significantly drop reproducibility owing to numerous steps and variations in protocols, resulting in higher error rates. This is a major stumbling block for clinical application of HTS, because there is not much room for error in the medical field, with people’s lives being at stake [95].

In summary, the most notable deterrents for clinical metagenomics are: (1) the complexity and high costs of sample preparation, (2) the requirement for special bioinformatic education and skills, (3) the need for a powerful data processing infrastructure and (4) the possible inconsistency between results owing to uneven pathogen distribution in a body and/or quality of sampling. Nevertheless, the mentioned limitations can potentially be alleviated with advances in sequencing technologies [96]. It is important to remember that nearly 40 years ago Sanger sequencing appeared cost-ineffective and overly complex for clinical application [95]; yet, it is one of the widely celebrated diagnostic tools in healthcare [97,98].

In most NGS-based pathogen studies, clutter reads are a problem. Whenever clinical samples are used, the presence of a host’s nucleic acids becomes an inevitability, requiring additional computational power for filtering them out and, consequently, adding to the gross sequencing cost. To make matters worse, target DNA/RNA usually constitutes only a small fraction of total DNA/RNA. However, target amplification combined with enrichment become a game-changer, allowing for both concentration of target templates and removal of unwanted sequences.

### 3.3. Methods for Improving Sequencing Output

Numerous methods are widely used: affinity enrichment [99], filtration [100], ultracentrifugation [101] and depletion of the host’s nucleic acids [90], including protocols utilizing saponin [102] and propidium monoazide [103]. Depletion is a powerful tool, but potential loss of target sequences restricts its use for samples with a low content of viral nucleic acids. Their fraction is usually lost during ethanol cleaning and stems from either weak probe affinity, competitive binding, low ethanol concentration or contamination with strong bases. Additionally, listed techniques are expensive and laborious. Unspecific amplification methods, e.g., multiple displacement amplification (MDA), utilize random primers and Phi29 polymerase to increase reaction output, however not necessarily with high enrichment quality [104], and may lead to additional contamination, uneven fragment distribution and amplification errors, lowering the overall library quality [105]. Sometimes depletion protocols do not effectively remove host-cell DNA [106]. Let us look further into specific techniques for improving NGS library concentration. (Figure 2)

#### 3.3.1. Nucleic Acids Depletion

Various types of clinical specimens routinely contain a large portion of host-cell nucleic acids, which can interfere with target PCR [18] and decrease sequencing efficiency and, therefore, have to be removed in a process called depletion. Utilizing biological, physical or chemical principles, it significantly decreases the content of host-cell DNA and/or RNA, while only negligibly lowering concentration of a pathogen’s nucleic acids. Most depletion kits are based on hybridization of molecular probes with target sequences, which is a lengthy but effective process. Commercial depletion kits are widely available (Ribo-Zero, RiboCop, HostZERO, NEBNext, MolYsis).

In a protocol by Gu et al. [107], Cas9 utilizes a set of guide RNAs to target unwanted nucleic acids during depletion of abundant sequences by hybridization (DASH). Cleavage-mediated depletion resulted in an up to 99% reduction of mitochondrial DNA. This approach can be used for targeting human genomic DNA in samples with viral nucleic acids.

Despite its beneficial role in sample processing, depletion poses a few risks for the output quality. Since it acts as a bottleneck, which allows only certain sequences through, biases are inherently imposed on the selection process. Cross-hybridization [108] is one example: recombinations in a viral genome or flaws in guide RNA design cause unspecific annealing to homologous non-target sequences, resulting in an erroneous cleavage.

Although separation of target sequences from “junk” nucleic acids is the primary goal of depletion, high levels of host-cell nucleic acids can, in theory, impede probe-sequence binding by competitive mechanisms, in which case the effectiveness of depletion would sag. However, some modified protocols have been developed that tackle this issue [103,109].

All points considered, nucleic acid depletion is an effective tool for decreasing concentrations of unwanted sequences in a sample, despite it possessing a few limitations that could lower the sequencing output by removing target nucleic acids.

#### 3.3.2. Hybridization-Based Enrichment

Experiments that compare modern clinical assays with HTS-based tests conclude that data derived from sequencing might be helpful in several ways. It can serve for pathogen detection and identification and also supply clinically valuable information on the presence of resistance genes or resistance-yielding mutations in a pathogen’s genome [110]. It could help healthcare professionals choose the correct therapy without having to wait for microbiological tests and prescribing broad-spectrum drugs that are constantly shown to have adverse side effects [111]. Of note is that NGS finds pathogens beyond standard testing panels, unlike traditional tests. However, genetic libraries must be created, often from samples with low concentrations of a pathogen’s nucleic acids and high content of nucleic acids from a host and other microorganisms. An elegant solution has been proposed by two independent teams of researchers, resulting in the development of a novel commercially available enrichment tool [112] based on hybridization.

The most notable feature of this approach is that molecular probes, instead of targeting selected viral species or families, bind to certain sequences in all known viral genera infecting vertebrates, thus supposedly creating the most comprehensive tool for viral screening. This enrichment technique greatly increases sequencing depth, while also raising the fraction of viral reads by 100–10,000 times compared to standard library preparation protocols [113,114]. A diagnostic tool of such sensitivity could serve as an alternative to traditional methods when they fail to detect the pathogen. In contrast, total sample sequencing could at least partly detect a new pathogen’s sequences, revealing its identity.

Target probe-based enrichment (also known as hybridization-based enrichment) is used for studying viral genomes without prior cultivation or clonal amplification [105,106,107,108,109,110,111,112,113,114,115,116,117]. For this, short DNA or RNA probes are used, which are complementary to the chosen sequences in a viral genome. The trick is to use several probes for one viral genome, thus ensuring its full coverage and guaranteed extraction: even if the genome contains major mutations, there is a distinct possibility that some target sequences remain unchanged, allowing for binding with a probe. Probes themselves are biotinylated at one end, binding strongly to beads covered in streptavidin. After that, a magnet separator is used to wash the beads from any redundant DNA and RNA with further elution of target sequences [118].

Hybridization-based enrichment holds a few advantages over PCR-based techniques. Alterations to a pathogen’s sequence impede primer annealing, whereas probes tend to bind to target sites, given proper reaction conditions and time, even if the target sequence contains polymorphisms [112]. Besides, hybridization works well for covering large genomes, demonstrating less coverage biases [119].

This approach has been successfully used to describe clinically significant viruses with varying genome sizes, such as HCV [54], HSV-1 [120], VZV [115], EBV [121], HCMV [122], HHV6 [119] and HHV7 [123]. Hybridization can be set up in a single tube and is easily automated [121]. Moreover, the use of probes yields results more reliable than those after cultivation. This is because acquired sequences are almost totally identical to original templates, whereas cultivation creates quasispecies, distorting actual genetic structure of a viral population [115,122].

Sensitivity of hybridization is increased when probes are designed using a large set of reference sequences, resulting in a better coverage of all conceivable genetic variants. This is achieved because target enrichment is still possible when target sequences differ from a probe sequence. However, full sequences are needed when creating hybridization probes, while designing PCR primers requires sequences only of the flanking regions. Another advantage of probe-based methods is that if some oligonucleotides fail to anneal during hybridization, overlapping probes might compensate [122,124].

Briese et al. [112] developed VirCapSeq-VERT—a platform for concentrating nucleic acids of vertebrate viruses. It contains approximately two million biotinylated probes for target enrichment. VirCapSeq-VERT detects virtually all human viruses, including new ones, featuring up to 40% of unknown sequences. Another team of researchers created an analogous approach, called ViroCap [117], which has been tested in a virological lab on actual clinical samples. In 26 samples, metagenomic shotgun sequencing (MSS) with ViroCap detected all anticipated viruses plus 30 that had not been identified before. An experiment carried out later [125] demonstrated that out of 30 viruses identified with NGS and missed by clinical tests, 18 were identified by MSS and ViroCap, and an additional 12 with ViroCap only. Another powerful computational tool for designing probes has been recently developed by Metsky et al. [126]. CATCH (compact aggregation of targets for comprehensive hybridization) analyzes target viral genomes and selects a minimum number of probes for optimal coverage of multiple viruses. This results in an effective hybridization-based enrichment, while preserving the original viral diversity of a sample. Reduction in the number of probes curtails experiment costs, while increasing the efficiency of metagenomic sequencing.

Nevertheless, the price per sample remains high for this approach, and it has not gained much popularity yet due to insufficient data on its relative performance. In addition, hybridization protocols are time-consuming. According to official protocols, hybridization may take up to 24 h in some cases, after which there are additional steps required for library preparation, consuming a few extra hours. Secondly, viral genomes that differ from probe sequences for more than 40% will most likely fail to undergo enrichment [112]. Although this does not guarantee the loss of these viruses, the effectiveness of the procedure nosedives. For these reasons, MSS and PCR with degenerate primers might prove more effective, albeit yielding more “noise” data in the output, which is removable during processing. In the end, the choice of enrichment method boils down to the experiment design and quality of the input material.

#### 3.3.3. Target Amplification

Target PCR is an effective and simple alternative to metagenomics, working well for both individual genes and small viral genomes [127]. PCR-based amplification is often used for whole viral genome sequencing of samples with low viral load [54], e.g., during the investigation of the measles outbreak during the Olympics in 2010 [128] and epidemics of Ebola [56] and Zika diseases [129]. Sequencing of long (2.5–3.0 kb) amplified fragments has clarified the variability of a *Norovirus* genome and its spreading among patients in a few hospitals in Vietnam [130,131]. A similar approach has been utilized to measure the specificity and sensitivity of the Illumina platform for detection of minor polymorphisms in mixed HIV populations [132]. Deep sequencing of PCR-amplified viral genomes yielded complete genome sequences for the influenza virus [133], Dengue virus [134] and HCV [135].

This approach proves apt for investigating small viral genomes that can be covered with only a few PCR amplicons. However, vast heterogeneity of RNA viruses (e.g., HCV27, *Noroviruses* and *Rhabdoviruses*) might require the use of multiple primer sets to ensure amplification of all known genotypes [130,136]. Some researchers propose that this method, coupled with NGS, is used for large viral genomes, like HCMV [19]. In this case, PCR allows long sequences to be “split” into shorter overlapping fragments that can be assembled into a full sequence during data analysis, which increases sequencing depth.

According to Reyes and Kim (1991) [19], another type of amplification is SISPA, a method that serves well when “nucleotide sequence of the desired molecule is both unknown and present in limited amounts making its recovery by standard cloning procedures technically difficult.” Recently, it has been used for quick and reliable identification and characterization of viruses [137].

As stated above, PCR works perfectly for amplification of small viral genomes, like those of HIV and influenza virus. For that purpose, primers are designed so that they cover a whole genome, either in a few fragments or in a single molecule. In theory, cloning shorter sequences is simpler, more reliable and increases sequencing depth, whilst also requiring compilation of primer panels for each genus. Nevertheless, target amplification has its own shortcomings. For instance, it has low data scalability, while requiring a relatively large amount of input material to ensure proper site coverage without biases. For this reason, its application is mainly restricted to samples that meet robust requirements (amount of input DNA, material quality, absence of PCR inhibitors, etc.). For instance, amplification of the whole Ebola virus genome utilizes 11 or 19 primer pairs, covering over 97% of the full sequence [138]. Two experiments studying *Noroviruses* included amplification with 14 and 22 primer pairs, respectively [137,138]. Finally, sequencing of the Paramurshir virus genome was conducted with a set of 60 PCR reactions and additional Sanger sequencing to cover the unamplified fragments [139].

Apparently, sometimes amplification, as described above, overloads laboratory workflow with numerous reactions and mandatory normalization of the products’ concentrations. There is always room for error, e.g., primers failing to anneal to targets due to unmatching primer sequences (particularly in rapidly mutating viruses [139]). It is also important to consider the costs of primer synthesis and reagents, along with the amount of labor required for setting up multiple reactions per each sample [140]. Thus, even though this method allows for amplification and further sequencing of large viral genomes, technical complexity and low cost-effectiveness render it inapt for massive clinical research, reserving it primarily for scientific purposes.

Multiple PCRs require more input material. In clinical practice, an amount of sample drawn from a patient is usually limited. Multiplex PCR allows for parallel amplification of target sequences in a single tube, thus utilizing a smaller sample volume. In this case, compatibility of primers has to be assessed beforehand using bioinformatic tools to avoid artifacts, such as primer-dimers and false priming.

Target PCR becomes a challenge when organisms with high genome mutability are studied, e.g., HCV [112], influenza virus or *Noroviruses*. Frequent changes in their genetic sequences interfere with primer annealing [54,130,141]. A more subtle approach to designing primers could improve the outcome, but PCR alone is by definition incapable of identifying new viruses.

We suppose that new methods based on metabarcoding might aid identification of viruses in the future. Originally, barcoding—another type of target amplification-mediated approach—was developed to assess bacterial associations (e.g., gut microbiota), based on reliable universal phylogenetic markers, such as 16S, where short taxon-specific sequences consisting of a variable fragment flanked by conservative fragments are used for tagging organisms. Metabarcoding combines a metagenomic approach (i.e., studying multiple organisms in a sample) with barcoding, making it a powerful tool for studying complex microbial associations [142], fungi and eukaryotes [143]. Usually, the scalability of a metabarcoding approach is limited by genus level; however, this limitation is compensated for by a significant cost reduction in comparison with metagenomics. Unfortunately, no such loci have been described for viruses, owing primarily to their genetic diversity; however, some attempts have been made at solving this issue with broad-range primer panels, created by computational methods [144,145] with promising results.

### 3.4. Whole Viral Genome Sequencing

Once a pathogen is identified, whole genome sequencing can be engaged for further research into its genome. This could prove beneficial for the knowledge of its genetic patterns, drug resistance and possible targets for antiviral therapy.

Comparatively small genomes, like those of human immunodeficiency virus (HIV), influenza virus, hepatitis B virus (HBV) and hepatitis C virus (HCV), are frequently sequenced in research purposes, and attempts are being made to integrate this procedure into clinical practice [146]. Until recently, detection of drug-resistant strains required sequencing of only certain parts of viral genome. Whole genome sequencing (WGS) was considered far too expensive for this purpose; however, we are obtaining more data on evolving mechanisms of drug resistance, including genetic mutations that define it [147], and with the cost of WGS steadily decreasing, the use of this method becomes justified.

Routine use of NGS for diagnostic purposes is expected to hold great advantages over existing techniques [148]. For instance, the genotyping of *Coronavirus* virus, which has been drawing attention of epidemiologists worldwide, is already influencing decision-making in healthcare [149]. Additionally, genomes of drug-resistant HIV strains are used to study the evolution of the virus [150] and complex genotype/phenotype associations between the virus and the host [151,152].

At present, WGS remains costly, even for small viral genomes, compared to the price of target sequencing. Nevertheless, additional data produced using WGS might supplement our knowledge of genetic substrates for drug resistance and uncover complex intergenic associations at its core.

### 3.5. Methods of Sequencing Data Analysis

Whenever metagenomics is used for the detection of pathogens, it is crucial to use reliable bioinformatic tools and specialized databases that would help decide whether discovered microorganisms indeed have caused the infection or act merely as artifacts. Such processing normally demands lots of computing power and knowledge and skills in bioinformatics. Typically, data analysis mainly involves comparing obtained reads with reference genomes. Quite a few algorithms have been developed for this purpose [153], with BLAST being the most widely used [154]. However, BLAST works slowly for the analysis of NGS data [155], and processing times can take several days, or even weeks, especially when calculations are performed for amino acids. Other often-used programs are Bowtie [156] and BWA (Burrows-Wheeler Aligner) [157], which are usually employed as filtration tools, and DIAMOND as an alternative to BLAST [158].

As for the published bioinformatic pipelines, in general they adapt either of the two strategies: (1) firsthand removing the host’s reads by mapping them to the host’s or another non-target reference genome and subsequently analyzing the remaining sequences; (2) assembling short reads into larger contigs, and only then comparing assembled sequences to reference genomes, including those of viruses [80]. (Figure 3)

First filtering of the host reads (Figure 3a) works well only when the host’s genome has been thoroughly studied and described in detail, although it is not a problem for detection of human viral infections. As HTS technologies advance, more complete genome sequences become publicly available. Another problem is that viruses may contain nucleotide fragments that are similar to certain regions in the human (or other host) genomes, leading to false negative results. The second approach requires high coverage to work, thus significantly increasing both the processing time and the experiment cost. Moreover, whenever input data are insufficient, spaces might form between reads, impairing their assembly into contigs, thus halting virus classification. In principal, two pipelines utilize the same concept of comparing sequences that “survived” filtering against reference viral genomes [80]. Depending on the sample type and expected contaminants, the filtering step may also include rRNA, mtRNA, mRNA, bacterial or fungal sequences or non-human host genomes.

Apart from the differences in the methodology between first filtering out the host reads or leaving them included, there are other serious challenges, such as the proper taxonomic assignment of viral reads or contigs. Homology or alignment-based similarity search methods are often used, but a composition search, in which oligonucleotide frequencies or k-mer counts are matched to reference sequences, is also an option. At the same time, a composition search requires the program to be “trained” on reference data and it is not used much in viral genomics.

Dismissing the host’s sequences is an important step that has been introduced into numerous processing algorithms, such as VirusFinder [77], VirusSeq [159] or Vy-PER [160]. This step helps remove false positives caused by similarities between some regions of the human and viral genomes. 

One popular pipeline for analysis of viral reads is by Petty et al. [161]—ezVIR, which was designed to process HTS data from any of the sequencing platforms, and which initially compares raw reads with reference human genome, subsequently removing them and analyzing the remaining sequences against the viral database.

VirusFinder, for instance, first utilizes Bowtie2 [156,162] to detect and remove sequences derived from human genome. Next, it engages BLAT (BLAST-like alignment tool) [163] to align the remaining sequences to the database of viral genomes. During the final step, the short reads, supposedly viral, are assembled into longer sequences—contigs—using Trinity [164]. VirusFinder, developed primarily to identify viral integration sites within the human genome, produces the best results when given sequencing reads with the maximum depth possible. In the original article [77], coverage varies between 31× and 121×. Acquired contigs are then used for phylogenetic analysis.

VirusHunter [165], on the other hand, utilizes BLASTn to filter out human-related sequences after quality evaluation. Sequences that pass filtration are taxonomically classified according to BLASTn and BLASTx algorithms. Thus, VirusHunter requires a high-quality host’s genome for sorting the sequences and significant computational power to run.

VirusSeq [159] is intended for detection of viral sequences in tumor tissues. First, it removes human sequences by comparing them against the reference. The MOSAIK program [166] is used both prior to filtration and afterwards to ensure quality sequence detection. VirusSeq sets limits on the minimally acceptable number of reads and coverage based on the size of viral genome. For example, it demands at least 1000 reads per virus given that the sequencing depth equals at least 30×. Although the threshold can be adjusted, this tool has been developed for reads with high coverage and is consequently not recommended for processing data with a low percentage of viral reads.

Vy-Per [160] is another bioinformatic instrument that utilizes reference human genome for filtering out reads of a host’s DNA. Sequences that are not dismissed during this step are compared with the data in the NCBI database using BLAT tool [163]. Although experiments set up to test Vy-Per use samples with a rather high coverage (80× for samples and 40× for controls), it is not mandatory, but lowers the risk of false positives.

PathSeq [167] is a powerful computational tool for analyzing the non-host part of sequencing data, which is able to detect the presence of both known and new pathogens. The PathSeq approach begins with a subtraction phase in which reads are aligned with human reference genome to subsequently exclude them, in an attempt to concentrate pathogen-derived data. This is followed by the analytical phase, in which the remaining reads are aligned to the microbial reference sequences and assembled de novo. The formation of large contigs, including several unmapped reads that do not have a significant similarity to alignment with any sequence in the referenced databases, may suggest a previously undetected organism.

SURPI (“sequence-based ultrarapid pathogen identification”) [168] is one more example of the pipelines for complex metagenomic NGS data generated from clinical samples that first filter out the non-target reads using 29 databases and then identify viral, bacterial, parasitic and fungal reads, which also involves de novo contig assembly.

There are many other tools available, and we refer to recent reviews by Nooij et al. (2018) [80] and Fonseca et al. [169].

The search for viral pathogens is often impeded by their genetic variability, caused by a multitude of factors: gene duplications and exchanges, frequent single-nucleotide mutations, gene-large insertions and rapid adaptation of viruses to new hosts. These become a particular nuisance whenever large sets of samples from various organisms are handled. Firstly, reliable reference genomes have only been assembled for a limited number of organisms, although, as previously mentioned, this issue is being addressed. Secondly, the amount of usable viral nucleic acids in a sample depends on numerous factors, including the stage of a pathogen’s life cycle and the overall quality of material and its type, all of which might cause prejudice towards certain viral species and drop the amount of usable viral DNA and RNA. This complicates the assembly of contigs, because it requires the maximum number of reads possible. 

All search methods rely on genome reference databases, such as the NCBI GenBank, RefSeq or BLAST nucleotide (nt) and non-redundant protein (nr) databases. The poor quality of the reference sequences is a major obstacle to processing data. Because of this, specialized databases are being compiled manually to ensure a strict quality control, e.g., ViPR [170], RVDB [171] and viruSITE [172]. However, they are reliable for the same reason that they are limited, because only a small fraction of all published sequences ever make it to these databases. As a consequence, reads obtained from supposedly new strains are frequently left out. Contrarily, a vast NCBI-based GenBank database is brimming with viral sequences, both complete and partial; however, the price for the quantity is the quality of the assembled data. Even so, GenBank is far from being truly comprehensive, but this is merely a question of time. Protein databases are also used, for example, Pfam [173] and UniProt [174]. Protein-level searches can usually detect more distant homology due to the usage of sensitive amino acid similarity matrices, which can improve the detection of divergent viruses, but untranslated regions of the genome remain unused.

## 4. Long Read Sequencing

Despite being one of the most powerful research tools, the NGS sequencing, which is sometimes called second-generation sequencing, is not flawless. Namely, not a single platform can read fragments more than 700 bp [175], thus necessitating DNA fragmentation prior to sequencing. Additionally, second-generation platforms amplify the signal prior to detection. PCR is used for molecular cloning of the fragmented DNA, which inevitably results in overrepresentation of some amplicons and uneven sequence coverage. Small length of the reads sometimes becomes a trouble for haplotyping, significantly complicating localization of the questioned loci. It also causes problems when analyzing repeats and recombinant regions of the DNA due to mapping issues.

In contrast, long read sequencing (LRS) technologies do not require significant DNA fragmentation or additional template amplification. For these reasons, they are frequently referred to as the “third-generation” sequencing technologies, although this distinction sometimes may not be accurate. Platforms such as PacBio (Pacific Biosciences), PromethION, GridION and MinION (ONT, Oxford Nanopore Technologies) are capable of reading exceptionally long fragments, including whole viral genomes. Secondly, these platforms potentially allow ultra-high coverage to be easily reached and even quantify the sample, because the viral load would be directly proportional to the number of detectable viral genomes in the sample. Thus, samples with high viral titer can provide high-quality sequences without prior amplification. Another feature of the ONT platforms is the sample turn-around time, which is often critical in the clinical environment; point-of-care testing makes it possible to identify the infectious agent virtually at the patient’s bed. In this setting, the correct therapy can be started within hours or days of disease manifestation, prospectively improving the patient care outcome.

One of the most popular ONT sequencers—the MinION—has been praised for possessing multiple boons: relatively low cost, when compared to platforms by Illumina and Thermo Fisher Scientific, and mobility, owing to compact dimensions of the sequencer, simplicity of protocols, and rapid sample processing. Another great advantage that ONT platforms hold over second-generation sequencing is the length of analyzed fragments. Because fragments over 2 Mb can be read in a single run with maintained quality, in metagenomic experiments, genes of interest, like those of drug resistance [176], can be easily attributed to relevant microorganisms, as opposed to reading shorter fragments, when identifying the resistant strains can, in theory, pose a significant challenge. Furthermore, available protocols include not only those for metagenomics, but also for studying select organisms, e.g., viruses. Both parallel and consecutive sequencing of samples can be performed, owing to a multitude of pores in a cell.

MinION has been extensively used in detecting known [177,178,179,180,181] and new [182] pathogen strains, for instance, of Ebola virus [183], *Begomovirus* [184] and *Papillomavirus* [185]. Curiously, MinION allows samples to be detected and genotyped within mere hours from sample collection, even “in the field”, sometimes literally [186,187]. It has also been successfully used for rapid investigation of Dengue virus outbreak in Angola [188] and Ebola epidemic in Guinea [189]. Another experiment [190] allowed for deciphering full genomes of three *Avipoxvirus* strains in real-time, skipping extraction and enrichment altogether. Another analogous example is Everglades Virus (EVEV), which has been detected and subgenotyped on-site [191].

Another great advantage of Nanopore technology is that unique protocols for direct RNA sequencing have been developed, meaning that RNA viruses could be analyzed right away, without reverse transcription [192,193]. Even more, depending on the experiment objective, either differential RNA sequencing (dRNA-Seq) or mRNA could be investigated, the latter being achieved by ligating an adapter only to the poly-A tail, thus allowing for transcriptome analysis [192,194]. Direct RNA-Seq helps avoid distortions of fragments’ representation [195] in the sample, revolutionizing the approach to studying RNA-viruses and the intricacies of their life cycle [196].

Despite the apparent benefits of this approach, there are a few drawbacks to consider, e.g., significant error rates that require a large coverage for their partial compensation. This poses a problem that is particularly important for the research of drug resistance, the substrate for which is often a few single-nucleotide polymorphisms, particularly in RNA-viruses [132]. Xu et al. suggest that incorrect sample differentiation and chimeric reads that emerge during sequencing with MinION might significantly alter the experiment outcome [197].

Combining target enrichment with long-range sequencing could potentially solve the problem of high error rate of the sequencing method [198,199] by increasing coverage and improving processing methods [200,201]. For instance, in one metagenomic study of sea water [202], the LASL method (long-read linker-amplified shotgun library) was used to produce enough DNA templates for successful MinION sequencing of a sample with initially low target DNA content (VirION system).

To summarize, small physical dimensions of MinION, along with other ONT sequencers, availability of DNA and RNA sequencing protocols, potentially high precision and high data processing rates create promising prospects for its clinical application [179]. So far, it has already been tested in veterinary science for detection of canine distemper virus (CDV) [203,204] and in biology for identification of numerous pathogens in plants [205] and fish [206]. Nevertheless, prior to extrapolating these positive results into the medical field, we need to develop and validate the workflows that would easily integrate into the everyday work of a clinical laboratory and not require any special bioinformatic training in the lab staff [207].

## 5. Obstacles to Overcome in the Nearest Future

The risk of contamination and the issues with sensitivity are important to sequencing viral nucleic acids, because they cause false positive and false negative results, respectively. High-sensitivity sequencing (be it metagenomic, PCR-based or probe-enriched) allows for detection of minimal quantities of contaminating viral nucleic acids [208,209]. For example, murine leukemia virus (MLV) [198,210] and *Parvovirus*-like sequences [99,211] are only two kinds of the vast range of common contaminants found even in commercial laboratory supplies [212].

Like with other highly sensitive technologies, only robust validated protocols and their strict implementation can effectively reduce the probability of contamination. This principle has been observed in action for quite a long time in paleogenetics [213], although methods in this field might prove somewhat exorbitant in everyday NGS research with samples not as valuable and unique. Nevertheless, some practices common in this field surely will find application in clinical NGS, when human wellbeing and life are at stake, compared to an abstract scientific interest of pure research.

It is also notable that detection of viral nucleic acids does not necessarily imply an ongoing infection, and therefore, positive results should be validated using supplementary methods targeted at pathogens suggested by the NGS tests. For instance, in the case of idiopathic encephalitis, positive results of NGS can be corroborated either using immunohistochemical assay [214,215], electron microscopy or microbiological methods, i.e., cultivation [216]. At any rate, setting validation thresholds and developing a method for NGS standardization is a significant challenge that has to be addressed prior to introducing NGS into clinical routine.

Undoubtedly, standardization of NGS-based methods, including data processing, will lay a foundation for their widespread integration into clinical routine. Existing algorithms for analyzing sequence output, given user-friendly graphical interfaces, should become easy to handle even without expert knowledge of Linux commands. Another important step is the creation of reliable, comprehensive drug-resistance databases that will enable detection of clinically significant strains required for the special treatment. Such databases are already available for HIV [217], HBV [218,219] and HCV [220].

Proper quality control is another problem. It is difficult to set a valid standard when the method is aimed at discovering new types of organisms. To confirm the NGS data, the pathogen in question has to be extracted, cultured as a pure strain and genetically characterized. Perhaps, the samples subject to NGS assays should be analyzed at several facilities to confirm the reproducibility of the results. The use of different sequencing platforms would ensure even better reproducibility and help prevent biases and typical mistakes that various sequencing technologies are prone to, while also excluding the human factor.

Despite multiple reasons clearly pointing to the necessity of NGS-based tests in virology, most importantly WGS, it is crucial to convince clinical laboratories that it is absolutely worth the effort. For this, clinicians should receive indisputable evidence of the advantages of this approach for the patient, institution budget and long-term prospects for the healthcare system. As for the laboratory staff, the workflow should be clearly structured, time-efficient, scalable and, preferably, automatable, rendering WGS a competitive approach with a highly convenient, reliable and useful outcome.

## 6. Conclusions

Studying viral genetics with NGS methods is rapidly gaining clinical significance, be it in diagnostics, epidemiological research, the hunt for drug resistance strains or infection control. Various approaches exist that feature WGS, amplicon sequencing, enrichment sequencing and metagenomics, the choice depending on the type of pathogen and experiment objective. Metagenomics, for instance, works best with unknown viruses, whereas PCR-based techniques are apt for samples with low diversity and short genomes. Target enrichment is preferred whenever viruses with large genomes are concerned or if the sample contains a multitude of heterogeneous viruses that feature well-identified nucleic acid sequences.

At present, two major research vectors exist: (1) developing techniques for depletion of bacterial and host sequences that would spare viral nucleic acids and (2) evolving long-range sequencing to the point where it becomes financially and qualitatively comparable to the second-generation methods. The key to success in the struggle against viruses and their shifty nature lies within combining current methods and thus potentiating the creation of the ultimate diagnostic tool. Ideally, it has to be able to detect and describe viral pathogens and predict the evolution of viruses to let clinicians combat them with the utmost effectiveness.

## Figures and Tables

**Figure 1 viruses-12-00211-f001:**
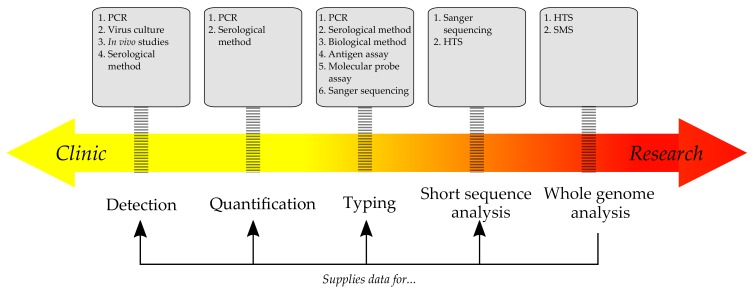
The prevalence of clinical or scientific application depends on the method and the type of data it yields. Classic approaches, like serology and PCR, are quick, but naturally limited to only the known pathogens. More advanced methods, such as HTS, could supply vital data for diagnostics (e.g., optimal target regions for PCR) and further clinical research.

**Figure 2 viruses-12-00211-f002:**
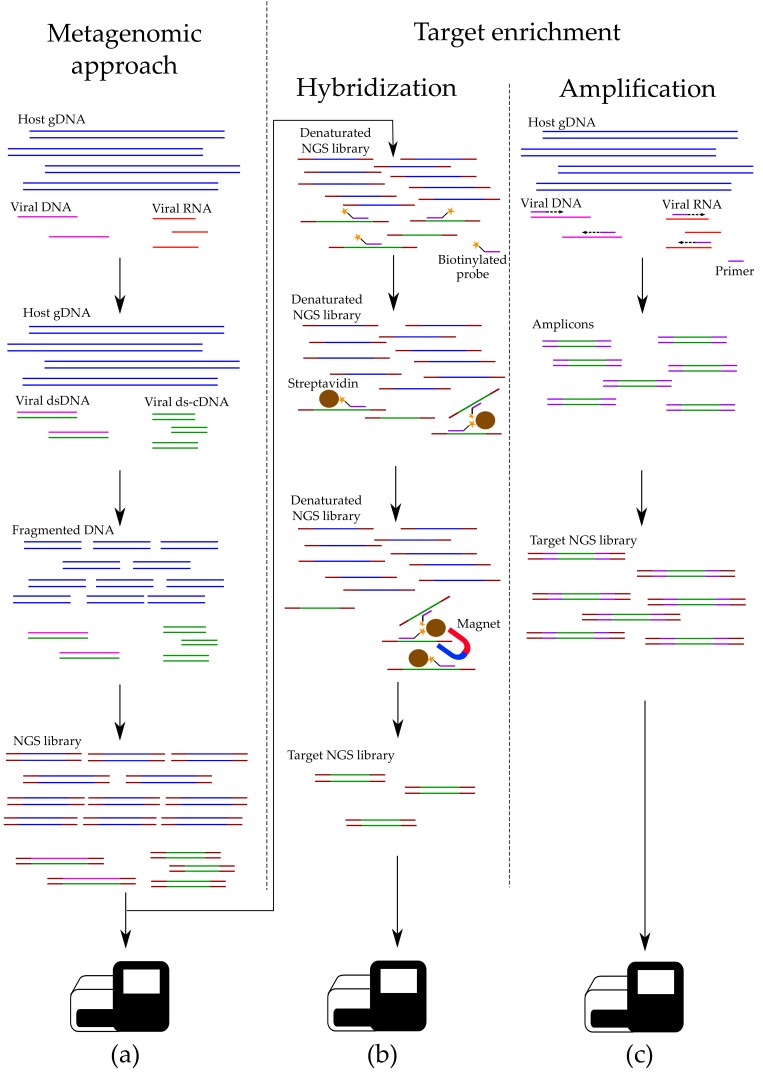
Unlike metagenomics (**a**), target HTS takes on an extra step of hybridization enrichment (**b**) and/or target amplification (**c**), during which the concentration of selected sequences is increased.

**Figure 3 viruses-12-00211-f003:**
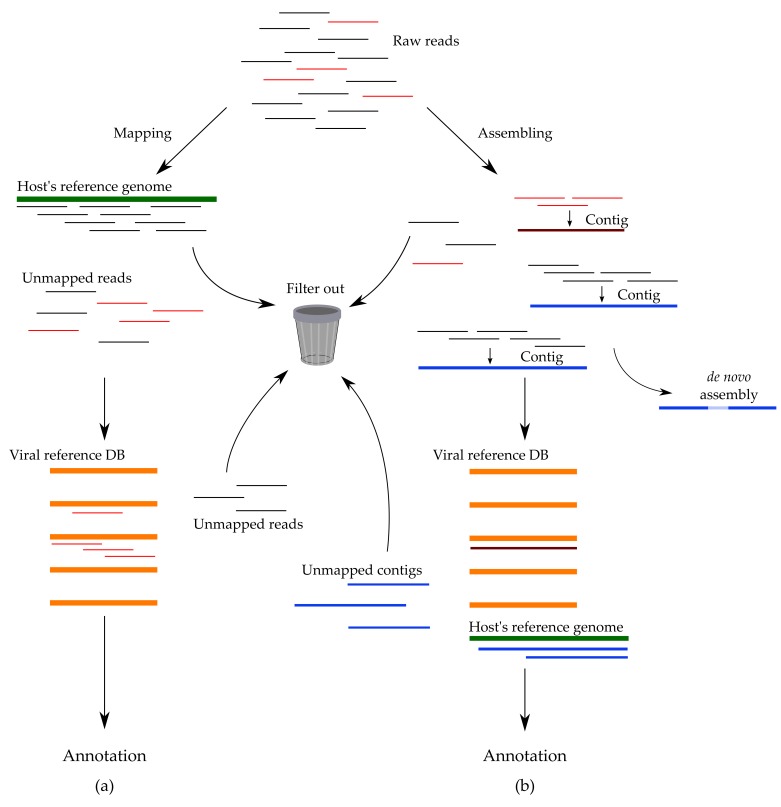
Pipelines for processing reads utilize two main strategies for filtering out “junk” data: (**a**) mapping raw reads onto the reference genome of the host and removing them, while preserving unmapped sequences for further analysis; (**b**) assembling short reads into contigs and comparing them against the host’s reference genome. Annotation follows both strategies.

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
