# Peer review of "Current Trends in Diagnostics of Viral Infections of Unknown Etiology"

_viruses, 2020, doi:10.3390/v12020211_

Round 1
Reviewer 1 Report
Kiselev et al. provide a review of both conventional and cutting-edge molecular techniques for the diagnosis of viral infections. The current revised manuscript is improved in several aspects, including a more balanced discussion of pros/cons, and a substantially expanded discussion of analysis tools. However, I still find the structure and writing to be redundant and somewhat confusing. I have provided some suggestions to address this below, but in general I think the manuscript would benefit greatly from further editorial intervention.
Section 1, Introduction
-This section still contains contradictions regarding the effectiveness of traditional diagnostic approaches; e.g. the second sentence says that “only a small fraction of clinical diagnoses are corroborated by the laboratory” (35-36), while the very next sentence says “in vitro diagnostic kits help identify the majority of common infections” (37-38). Clearly, these statements cannot both be true, and it reduces the readers’ enthusiasm for and trust in the paper to have this kind of discrepancy so early on. There are references that the authors could include to define exactly what percentage of infections go undiagnosed for specific syndromes such as central nervous system infection.
Section 2, Traditional methods of diagnosing infections
- What technique is being described in lines 241-248?
- The added description of real-time PCR is helpful but should come before broad-range PCR and also highlight some specific viral PCR tests that are in clinical use (enterovirus, influenza, herpes simplex virus, etc).
Section 3, studying viral pathogens with high-throughput sequencing
- I don’t think enterovirus D68 is a good example of the benefits of next-generation sequencing (Line 731), since it is currently suspected to be an important new cause of acute flaccid myelitis, but is generally not detectable by sequencing. There are other examples that could be used for identification of novel viruses
- The efficacy of metagenomic sequencing in laboratory diagnostics has been assessed in several recent papers (in contrast to line 749), including several that the authors have in their references
- In general, I found the organization of this section to be very confusing. I would recommend keeping sections 3.1 and 3.2 in their current order, then move directly to the sections about hybridization-based enrichment and nucleic acid depletion (since they address some of the problems the authors raise in section 3.2).
- I found Section 3.4 to be especially out of place. The title of this section makes it sounds like it’s going to be about endogenous retroviruses or other viruses “in” the human genome, but most of it is not.
- Section 3.5.1 – it seems like this should come after the target enrichment and depletion, since amplification is an alternative whereas the other two are improvement methods, as stated in the 3.5 intro section (lines 1318-1337)
- I don’t really understand what the authors mean by viral DNA metabarcoding (line 1725)
Section 4, Third-generation sequencing
- I’m not sure that the distinction between second generation and third generation sequencing is necessary or accurate, since long-read methods e.g. PacBio have been around for a decade. It might be more clear to describe the distinction as short-read versus long-read sequencing.
- It is confusing to interchange “ONT” and “MinION” (lines 2377-2385)
- In general, this section could be substantially condensed
- The parts about target enrichment and host nucleic acid depletion in this section are redundant with earlier sections and could be moved/eliminated
Author Response
Thank you for your invaluable input on our article. We have carefully reviewed your comments and adjusted the text and the figures accordingly. Our response is given below in a point-to-point manner.
Section 1, Introduction
-This section still contains contradictions regarding the effectiveness of traditional diagnostic approaches; e.g. the second sentence says that “only a small fraction of clinical diagnoses are corroborated by the laboratory” (35-36), while the very next sentence says “in vitro diagnostic kits help identify the majority of common infections” (37-38). Clearly, these statements cannot both be true, and it reduces the readers’ enthusiasm for and trust in the paper to have this kind of discrepancy so early on. There are references that the authors could include to define exactly what percentage of infections go undiagnosed for specific syndromes such as central nervous system infection.
Your comments made it clear that the introduction required major editing, so we have rewritten the entire section to ensure cohesion between the statements, as well as to state our main points in a more appealing way, so that the reader could deduce more easily what we are going to review in the paper. We have also included some statistics with references to corroborate our points (lines 32-73).
Section 2, Traditional methods of diagnosing infections
-What technique is being described in lines 241-248?
We have attempted to describe a theoretical method based on broad-range PCR that could aid the research of viral genome, however, after some consideration we have decided that similar ideas are discussed down the line, and therefore have removed this paragraph.
-The added description of real-time PCR is helpful but should come before broad-range PCR and also highlight some specific viral PCR tests that are in clinical use (enterovirus, influenza, herpes simplex virus, etc).
We have added a brief comment on available virus-specific kits (lines 119-123).
Section 3, studying viral pathogens with high-throughput sequencing
- I don’t think enterovirus D68 is a good example of the benefits of next-generation sequencing (Line 731), since it is currently suspected to be an important new cause of acute flaccid myelitis, but is generally not detectable by sequencing. There are other examples that could be used for identification of novel viruses
Instead of enterovirus D68, we have mentioned an Arenavirus case with a relevant reference (lines 171-173).
- The efficacy of metagenomic sequencing in laboratory diagnostics has been assessed in several recent papers (in contrast to line 749), including several that the authors have in their references
We are grateful for this remark. Indeed, numerous papers investigated this issue, which aptly falls into the scope of the current paper, so we have included some of them as references (line 185).
- In general, I found the organization of this section to be very confusing. I would recommend keeping sections 3.1 and 3.2 in their current order, then move directly to the sections about hybridization-based enrichment and nucleic acid depletion (since they address some of the problems the authors raise in section 3.2).
In accordance with your recommendations, as well as suggestions made by other reviewers, we have changed the structure of this paper. Namely, sections elaborating on depletion and hybridization have been moved down the line of sections about metagenomics. We introduce the section on depletion before enrichment, because it seems to reflect the logic of sample processing.
- I found Section 3.4 to be especially out of place. The title of this section makes it sounds like it’s going to be about endogenous retroviruses or other viruses “in” the human genome, but most of it is not.
Indeed, after reviewing this part we have concluded that it fails to fit into the paper structure, so it has been removed.
- Section 3.5.1 – it seems like this should come after the target enrichment and depletion, since amplification is an alternative whereas the other two are improvement methods, as stated in the 3.5 intro section (lines 1318-1337)
As we have mentioned previously, considerable alterations to the paper structure have been made, resolving this issue, to the best of our judgement.
- I don’t really understand what the authors mean by viral DNA metabarcoding (line 1725)
Because barcoding and metabarcoding (references included in the text) have not been confidently applied to virological research, we have decided to merely sketch out its possible future use, and therefore restructured and shortened this section, moving it to the section on target amplification, because, in essence, it is a special case of TA (lines 454-465).
Section 4, Third-generation sequencing
- I’m not sure that the distinction between second generation and third generation sequencing is necessary or accurate, since long-read methods e.g. PacBio have been around for a decade. It might be more clear to describe the distinction as short-read versus long-read sequencing.
We have elaborated on the differences between the short-read and the long-read sequencing (lines 580-612). Concerning the term “third generation sequencing”, we agree that the term is neither informative nor strictly accurate; however, we mention it because it is frequently used in scientific literature, perhaps, slightly emotionally, to underscore the vast distinction between SR and LR sequencing.
- It is confusing to interchange “ONT” and “MinION” (lines 2377-2385)
We have corrected this mistake and separated the name of the technology from the name of the platform.
- In general, this section could be substantially condensed
During the previous review, we received a comment regarding that this section did not contain enough data on the subject, so we expanded it. We have now cropped the text to retain merely the most important information on long read sequencing and its clinical application.
- The parts about target enrichment and host nucleic acid depletion in this section are redundant with earlier sections and could be moved/eliminated
As we have mentioned previously, considerable alterations to the paper structure have been made, resolving this issue, to the best of our judgement.
Reviewer 2 Report
General comments
The subject of the review is topical and the technology likely to have a significant impact in the field of clinical diagnostics. There is therefor likely to be wide appeal. The general standard of writing is acceptable but further proofreading and editing for clear scientific language is desirable. For example, informal expressions such as ‘on the bright side’ and ‘gleaming abundance’ should be rephrased to more formal language. While not containing any obvious errors, too often the manuscript reads as a list of short statements with associated references. Some of the issues raised are explored at great length while others are summarised in a single line or paragraph, with no obvious reason why certain topics are preferred. There could be some value for a reader relatively new to the field but the review does not provide a particularly deep analysis of the subject and there are no novel insights. The authors are encouraged to review the manuscript and ensure a more balanced discussion of the various topics and clear summaries/insights for each area.
Specific comments
Certain aspects of the introduction are not relevant to the subject and should be removed. For example, the discussion of therapeutics for and factors affecting the increasing spread of emerging viruses are beyond the scope indicated by the title and abstract – the article would be improved by focusing just on the diagnostic aspects. Line 39: References should be added to support the assertion that “… the number of infectious with non-specific symptoms has been steadily increasing…” as this is an important point in the argument for improved diagnostics. Line 71: The term ‘soft thermodynamic conditions’ is confusing. I am not sure what the intended meaning is, but if this refers to PCR then perhaps it would be preferable to simply say ‘… by polymerase chain reaction’ Line 93: The term ‘… not quite sensitive.. ‘ would be better phrased as ‘..not sufficiently sensitive..’. Line 100: It is assumed this sentence refers to the challenge in multiplexing of different sets of primers and probes. If so, it is suggested that the phrase ‘multiplex’ be included in the sentence as readers are likely to be more familiar with this term. It could also be mentioned about the limits on the number of targets that can be analysed in a single reaction due to the number of fluorophores available and that having to run a greater number of reactions impacts on sample volume requirements. Line 138-139: The assertion that there is ‘… no prejudice specific organisms.’ is questionable. It is becoming clearer that certain biases exist that affect the relative proportions of different virus structures and genomes. The authors should mention the potential biases due to high GC% genomes, the resistance to extraction of some small non-enveloped viruses and the effects of the physical structure of the genome on its amplification efficiency. An example reference is Wilcox, Delwart and Diaz-Munoz, 2019, Microbial Genomics. The authors are encouraged to consult the literature for other appropriate references. Line 140: there is one other major methodological limitation – sensitivity and how to ensure that the level of sensitivity is known and adequate for every sample. There is brief mention of standards in line 733 but significant further discussion is merited as this is one of the major challenges to be addressed. Section 2 discusses molecular methods at length, which is appropriate given their prevalence in diagnostic procedures. However immunoassays are mentioned only in passing. It would be useful to have a paragraph outlining the strengths and limitations of immunoassays in diagnostics. As mentioned in general comments, it is not clear why certain topics are explored in great detail and others only in passing. The discussion of ONP is disproportionately long for what is still a very new product. While it undoubtedly has potential, it is some distance from routine application and comes nowhere near some of the second generation systems in terms of quality and depth. This section should be significantly reduced in length. Line 224 and general: The point about evaluating the minimum SRPS is correct and this is a critical point in the development of HTS to the point where it can possibly be used for routine diagnostics. Given the likely high and variable quantity of host-cell nucleic acid and the fact that metagenomics measures the relative rather than absolute levels of nucleic acid, it is challenging to define an appropriate threshold for assigning a positive or negative result to a sample. For pathogens at high concentration this may not be a problem but at low concentrations it will not be clear whether a negative result simply reflects a low relative concentration of pathogen nucleic acid. This issue needs to be addressed in the manuscript – no definite solution has been achieved but several research papers and perspectives articles have been published on the topic. An example reference that could be consulted is Khan et al…mSphere. 2017 Sep 13;2(5). doi: 10.1128/mSphere.00307-17. Line 256: the issues described are not in fact related to the sequencing technology, so strictly speaking improvements in sequencing won’t address them, though they may compensate for them in some cases. Line 405: earlier in the paper the difficulties and limitations of multiplex primer sets are highlighted yet here it is presented as a possible solution. There should be a consistent message. Line 427: the same limitations will apply to HTS, or indeed any detection method. Line 474: Nucleic acid depletion steps can introduce bias. This should be discussed. Particularly in the case of novel virus discovery where the effect on virus nucleic acids will be even more uncertain. Lines 721-723: It is correct to state that nucleic acid based tests do not confirm ongoing infection, however the comment about validating results using supplementary methods applies equally to PCR or immunoassays, both of which are often used as a definitive diagnostic assay. Conclusion: as mentioned previously, the issue of setting appropriate thresholds for positive signals should be highlighted as one of the major areas for research if HTS is going to be successfully and widely applied in clinical diagnostics.Author Response
Thank you for your invaluable input on our article. We have carefully reviewed your comments and adjusted the text and the figures accordingly. Our response is given below in a point-to-point manner.
Specific comments
Certain aspects of the introduction are not relevant to the subject and should be removed. For example, the discussion of therapeutics for and factors affecting the increasing spread of emerging viruses are beyond the scope indicated by the title and abstract – the article would be improved by focusing just on the diagnostic aspects. Line 39: References should be added to support the assertion that “… the number of infectious with non-specific symptoms has been steadily increasing…” as this is an important point in the argument for improved diagnostics.
Your comments made it clear that the introduction required major editing, so we have rewritten the entire section to ensure cohesion between the statements, as well as to state our main points in a more appealing way, so that the reader could deduce more easily what we are going to review in the paper. We have also included some statistics with references to corroborate our points (lines 32-73).
Line 71: The term ‘soft thermodynamic conditions’ is confusing. I am not sure what the intended meaning is, but if this refers to PCR then perhaps it would be preferable to simply say ‘… by polymerase chain reaction’
We have rephrased this sentence accordingly (line 97).
Line 93: The term ‘… not quite sensitive.. ‘ would be better phrased as ‘..not sufficiently sensitive..’.
We have rewritten this section, removing the sentence (lines 116-128).
Line 100: It is assumed this sentence refers to the challenge in multiplexing of different sets of primers and probes. If so, it is suggested that the phrase ‘multiplex’ be included in the sentence as readers are likely to be more familiar with this term. It could also be mentioned about the limits on the number of targets that can be analysed in a single reaction due to the number of fluorophores available and that having to run a greater number of reactions impacts on sample volume requirements.
We have included the suggested information into the text and have partly rewritten it. We have also included the term “multiplex. ” (lines 125-128)
Line 138-139: The assertion that there is ‘… no prejudice specific organisms.’ is questionable. It is becoming clearer that certain biases exist that affect the relative proportions of different virus structures and genomes. The authors should mention the potential biases due to high GC% genomes, the resistance to extraction of some small non-enveloped viruses and the effects of the physical structure of the genome on its amplification efficiency. An example reference is Wilcox, Delwart and Diaz-Munoz, 2019, Microbial Genomics. The authors are encouraged to consult the literature for other appropriate references.
We have added extra information regarding the biases that affect the method. Thank you for this valuable input and the useful reference (lines 155-167).
Line 140: there is one other major methodological limitation – sensitivity and how to ensure that the level of sensitivity is known and adequate for every sample. There is brief mention of standards in line 733 but significant further discussion is merited as this is one of the major challenges to be addressed.
We have extended this section to cover more on the issue (lines 264-276).
Section 2 discusses molecular methods at length, which is appropriate given their prevalence in diagnostic procedures. However immunoassays are mentioned only in passing. It would be useful to have a paragraph outlining the strengths and limitations of immunoassays in diagnostics.
We have added two paragraphs discussing immunoassays in a way that we hope our readers would find helpful, while retaining brevity (lines 75-94).
As mentioned in general comments, it is not clear why certain topics are explored in great detail and others only in passing. The discussion of ONP is disproportionately long for what is still a very new product. While it undoubtedly has potential, it is some distance from routine application and comes nowhere near some of the second generation systems in terms of quality and depth. This section should be significantly reduced in length.
In accordance with previous comments and your remark, we have adjusted the size of this section.
Line 224 and general: The point about evaluating the minimum SRPS is correct and this is a critical point in the development of HTS to the point where it can possibly be used for routine diagnostics. Given the likely high and variable quantity of host-cell nucleic acid and the fact that metagenomics measures the relative rather than absolute levels of nucleic acid, it is challenging to define an appropriate threshold for assigning a positive or negative result to a sample. For pathogens at high concentration this may not be a problem but at low concentrations it will not be clear whether a negative result simply reflects a low relative concentration of pathogen nucleic acid. This issue needs to be addressed in the manuscript – no definite solution has been achieved but several research papers and perspectives articles have been published on the topic. An example reference that could be consulted is Khan et al…mSphere. 2017 Sep 13;2(5). doi: 10.1128/mSphere.00307-17.
We have extended this paragraph with extra data and discussion (lines 234-248).
Line 256: the issues described are not in fact related to the sequencing technology, so strictly speaking improvements in sequencing won’t address them, though they may compensate for them in some cases.
We have rephrased the sentence accordingly (lines 284-286).
Line 405: earlier in the paper the difficulties and limitations of multiplex primer sets are highlighted yet here it is presented as a possible solution. There should be a consistent message.
We have outlined the disadvantages that greatly limit the application of target amplification, noting that it might work well in some cases to improve the sequencing output, however, not in all cases (lines 424-453)/
Line 474: Nucleic acid depletion steps can introduce bias. This should be discussed. Particularly in the case of novel virus discovery where the effect on virus nucleic acids will be even more uncertain.
We have added the indicated data (lines 323-327).
Lines 721-723: It is correct to state that nucleic acid based tests do not confirm ongoing infection, however the comment about validating results using supplementary methods applies equally to PCR or immunoassays, both of which are often used as a definitive diagnostic assay. Conclusion: as mentioned previously, the issue of setting appropriate thresholds for positive signals should be highlighted as one of the major areas for research if HTS is going to be successfully and widely applied in clinical diagnostics.
We have added information on standardization issues, including validation (lines 264-278).
Reviewer 3 Report
Daniel and colleagues summarize the current trends of diagnostic virology and present the potential values of high-throughput sequencing (HTS) or next-generation sequencing (NGS) against viral infection. By comparing to the conventional clinical tests, the authors demonstrate HTS/NGS will be more universal and precise in diagnostics of viral infections if its pipelines and costs can be standardized and reduced in the future. In the manuscript, these authors do not only introduce the advantages of HTS/NGS in diagnostic virology but also describe the problems that should be improved to facilitate scientific and clinical research. They provide sufficient evidence by citing lots of references. However, I list several specific comments below for the authors’ consideration.
Page 3, Figure 1. Please indicate the meaning of the black arrows in this figure. In my opinion, whole genome sequencing analysis can be used in all of these clinical and scientific applications, but not limited to “Detection” and “Typing”.
Page 3, Part 3 (start from line 117). Can the authors briefly introduce the basic concept of “Metagenomics” at the beginning to make the section clear?
Part 3.3, 3.4, and 3.5 seem very confusing in this manuscript. I advise the authors to reorganize these parts and describe the features of metagenomics with clarity and logic.
Author Response
Thank you for your invaluable input on our article. We have carefully reviewed your comments and adjusted the text and the figures accordingly. Our response is given below in a point-to-point manner.
Page 3, Figure 1. Please indicate the meaning of the black arrows in this figure. In my opinion, whole genome sequencing analysis can be used in all of these clinical and scientific applications, but not limited to “Detection” and “Typing”.
We have changed the figure accordingly.
Page 3, Part 3 (start from line 117). Can the authors briefly introduce the basic concept of “Metagenomics” at the beginning to make the section clear?
We have extended this section to explain the basic concept of this approach (lines 146-153).
Part 3.3, 3.4, and 3.5 seem very confusing in this manuscript. I advise the authors to reorganize these parts and describe the features of metagenomics with clarity and logic.
In accordance with your recommendations, as well as suggestions made by other reviewers, we have changed the structure of this paper. Namely, sections elaborating on depletion and hybridization have been moved down the line of sections about metagenomics. We introduce the section on depletion before enrichment, because it seems to reflect the logic of sample processing.
Round 2
Reviewer 1 Report
The authors have provided a second revision to their review of diagnostics for viral infections. Overall, the introduction is stronger and the structure now seems more logical. I think this would be a beneficial resource to readers of Viruses with a few additional changes.
Although improved from before, I am not sure that section 2 (traditional methods) adds much to the paper, especially the section on immunoassays – would consider removing or considerably shortening this. The references do not seem to line up Line 246 – real-time PCR is commonly used in clinical diagnostic laboratories, so if the authors state that the view is “open for debate”, please provide references to support this perspective. I am not sure that the term “clutter DNA” will be widely understood For hybridization techniques, please also see Metsky et al. Nature Biotechnology 2019Author Response
Thank you for your invaluable input on our article. We have carefully reviewed your comments and adjusted the text and the figures accordingly. Our response is given below in a point-to-point manner.
Although improved from before, I am not sure that section 2 (traditional methods) adds much to the paper, especially the section on immunoassays – would consider removing or considerably shortening this.
We have removed the brief description of the method’s history, as well as some basic description of Sanger sequencing workflow downstream. We also focused the discussion on benefits and drawbacks of each conventional method, thus shortening the section (lines 75-121).
The references do not seem to line up Line 246 – real-time PCR is commonly used in clinical diagnostic laboratories, so if the authors state that the view is “open for debate”, please provide references to support this perspective.
PCR is indeed widely deemed a gold standard for diagnostics of infections diseases, so we have corrected this sentence to clearly state this point (line 107).
I am not sure that the term “clutter DNA” will be widely understood
We agree that this turn can be interpreted in several ways, and have therefore changed it to clarify that we are, in fact, talking about host-cell DNA.
For hybridization techniques, please also see Metsky et al. Nature Biotechnology 2019.
Thank you for suggesting this article. We have included a section on CATCH into our review, because a brief overview of this cutting-edge technology would indeed be beneficial for our readers (lines 376-380).
Reviewer 2 Report
General comments
The standard of the writing has slightly improved. There are still some sections that don’t read smoothly however it is of adequate quality. My original comment regarding the lack of deep analysis or novel insights still stands. While not containing obvious errors, there is still doubt about the value of the review. Nevertheless, there may be some use to readers new to the field.Specific comments
Lines 104- 113 – While it was requested to add some discussion about immunoassays, the discussion of ELISA is unnecessarily long. The techniques is well established and the history of its development is not required for this review. Discussion should centre around the relative strengths and limitations of the technique and how it compares to other diagnostic techniques. Line 442-443 – the sentence about ‘unbiased sequencing’ is immediately contradicted in the following discussion of the various biases known to exist within metagenomics. There should be consistent across the manuscript. Lines 1579 -1686 – the discussion of ONT has not been appreciably condensed and is still disproportionate to the relative utility of this technology compared to tried-and-tested platforms such as Illumina and IonTorrent.Author Response
Lines 104- 113 – While it was requested to add some discussion about immunoassays, the discussion of ELISA is unnecessarily long. The techniques is well established and the history of its development is not required for this review. Discussion should centre around the relative strengths and limitations of the technique and how it compares to other diagnostic techniques.
In accordance with your suggestions, as well as suggestions from another reviewer, we have curtailed this section, hopefully removing the fragments you might find redundant (lines 75-121).
Line 442-443 – the sentence about ‘unbiased sequencing’ is immediately contradicted in the following discussion of the various biases known to exist within metagenomics. There should be consistent across the manuscript.
Indeed, we have overlooked this apparent contradiction. We meant to say that metagenomics merely had the potential of unbiased sequencing, which does not always coincide with actuality. We have therefore changed this sentence in accordance with your suggestion (lines 145-147).
Lines 1579 -1686 – the discussion of ONT has not been appreciably condensed and is still disproportionate to the relative utility of this technology compared to tried-and-tested platforms such as Illumina and IonTorrent.
We are excited about the ways in which ONT could aid diagnostics of viral infections, which is why we have attempted to explore it in detail, obviously going too great a length. Thank you for pointing this out. We have edited the section, removing a paragraph on VolTRAX and custom protocols altogether, as they are out of the scope of the review and can be omitted. We hope that you will find the current version of this section properly condensed.
This manuscript is a resubmission of an earlier submission. The following is a list of the peer review reports and author responses from that submission.
Round 1
Reviewer 1 Report
This is a very comprehensive review of the "current trends in diagnostics of viral infections of unknown etiology". The authors cover many aspects extensively, sometimes maybe even a bit too lengthy. Shortening some paragraphs and omitting some examples would maybe help to make the review more focused (e.g. less examples of pipelines lines 529–554).
# Major comments
- I somewhat disagree with calling target amplification by PCR (Figure 2c) an enrichment process. Enrichment is for me rather a process like hybridization capture (Figure 2b) where no amplification takes places. I would therefore suggest changing the panel titles in Figure 2 and also the text accordingly.
- Apart from enrichment also depletion of unwanted sequences is an option (e.g. RiboZero, methylated DNA). These possibilities are mentioned in the third generation sequencing part (line 670) but I would move them further up in the text.
- There are lot of examples of cases/studies using targeted amplification (Zika, Ebola, CDV, HIV, …); however, for targeted amplification the specific target sequence has to be known and this does therefore not really match the scope of the review of diagnostics of unknow etiology.
- I somehow miss the aspect of quality controls and validation of such assays, especially as the targets are be unknown. Maybe the authors could add a sentence or two (external quality controls, reagents for quality control).
# Minor comments:
- line 21: Why here "NGS" while otherwise "HTS"? "Next Generation Sequencing" is singular, however, the authors use plural when referring to NGS
- line 50: What has antibiotic resistance to do with emerging viruses? Please explain or add a reference.
- line 214: I assume 10^6 reads/min (not 106 reads/min).
- line 274: Irreproducibility of the results: How do the authors come to this conclusion? Please explain or add a reference.
- line 283: Resistance testing e.g. for HIV still needs only sequencing of particular parts of the genome, irrespective whether it's done by NGS or Sanger Sequencing.
- line 334: For host depletion I suggest adding some more citations, maybe depletion with saponin as described by Charalampous, T. et al. Nanopore metagenomics enables rapid clinical diagnosis of bacterial lower respiratory infection. Nat Biotechnol 37, 783–792 (2019).
- line 341: What is bric‐a‐brac DNA? I know there is a bab locus? Why is this important?
- line 352: The reference is not formatted correctly.
- line 529: Reference 150 is not Petty et al, but Krupp et al. Why is the pipeline by Petty one of the most popular, but then it's not discussed in following paragraphs?
Reviewer 2 Report
Kiselev et al. provide a review of both conventional and cutting-edge molecular techniques for the diagnosis of viral infections. Although this is a very important and timely topic, unfortunately I do not think the review as currently written will help readers understand this complex and rapidly-changing field. In general, I found the review to be written in a somewhat confusing and redundant manner, and several areas contained inaccuracies or statements that I think are too strongly worded for this nuanced topic. I have noted some of these below but would encourage the authors to revise the entire manuscript carefully to address the broader points noted above.
Major points:
1) I disagree with the following statements (among others) and would encourage the authors to revise and/or provide references to support these statements:
Lines 20-22: “Compared to conventional clinical tests, NGS are substantially more informative, precise, and are less prone to errors.” (the authors themselves contradict this on lines 267-270)
Lines 38-39: “the majority of infections that are common in the middle latitudes possess a readily identifiable clinical presentation with pathognomonic diagnostic signs”
Lines 73-88, describing Sanger sequencing. This technique is most often performed by first amplifying a specified region, not “amplification of the total DNA” (line 75); because of this, the levels of host DNA (line 81) are not generally a problem. Filtration and exonuclease treatment (lines 83-85) can be used upstream of any technique, and are not specific to Sanger sequencing.
Lines 183-184: “metagenomics show a substantial potential for differentiating possible causative agents from actual etiological factors of infection.” Metagenomics allows detection of any microbe, whether pathogenic or not.
2) There are a large number of metagenomic classification tools available, and it would benefit readers to be aware of this, rather than just Taxonomer (lines 212-214); if the authors do not want to include this discussion in this paper, they could refer readers to one of several excellent recent reviews.
3) In the discussion of barcoding (lines 406-432), I would very clearly distinguish between barcodes that are routinely added to libraries to allow multiplexing prior to sequencing, from barcodes that are used to identify individual RNA molecules / viral genomes.
Reviewer 3 Report
This manuscript gives a comprehensive review detailing NGS techniques and their application in. viral diagnostics. The authors provide a nice overview of more traditional techniques. They further give a detailed summary of different approaches and enrichment techniques that can applied for the identification of viral pathogens by massive parallel sequencing. In addition, a relatively brief overview of bioinformatic tools used for data analysis is given. Finally, the application of third-generation long-read sequencing technology for viral diagnostics is discussed and the limitations of current techniques are highlighted.
While this is an interesting and detailed review, I do have some remarks.
Major remarks:
-The language throughout the whole text should be corrected by a native English speaker.
-Throughout the review a distinction between different sample matrices is not discussed. Viral load and viral to host nucleic acid ratio can be highly variable and have a high impact on the success rate and methodology used. This should be discussed
-While there is a difference in the methodology between first filtering out the host reads or leaving them included, it is a bit presumptive to identify this as the only important methodological difference in analysis pipelines. One of the main challenges in virus discovery is the proper taxonomic assignment of viral reads/contigs. The differences in methods (k-mer based, homology based, sequence composition based, hybrid methods, marker gene based) and most commonly used search algorithms (BLAST variants, hidden markov models, alignment-based methods) could be highlighted. ONT technology has several benefits for diagnostics. Not all viral sequence analysis tools are compatible with long read sequencing. Examples of long read data analysis tools will be of added value here.
-It is of added value to discuss other forms of enrichment (e.g. virus culture, ultracentrifugation, enzymatic digestion of host nucleic acids) in greater detail
Minor remarks
Line 21: NGS needs to be written in full
Line 43 Flavivirus, genus names should be italicized
Line 44 Alphavirus, genus names should be italicized
Line 83-84; this line is not clear; filtration does not always entail subsequent ultracentrifugation
Line 80-88 references should be included
Line 107-108 When properly designed, RT-PCR remains the golden standard for diagnosis and follow-up of viral infections. Could you please rephrase this?
Line 109-112 Commercial, syndromic panels are becoming more and more a staple in clinical diagnostics. These panels include all major expected causative pathogens and can provide a quick diagnosis in most cases, although they come at a significant cost. For the unusual suspects the authors are correct to point out NGS can be of significant added value. This should be rephrased.
Figure 1.
Science should be changed to Research
Cultural method should be changed to virus culture
Biological methods is a vague term and should be replaced.
Antigen and molecular probe assays should be included.
ELISA is an example of a serological method and should not be a separate bulletin.
Line 143 an ELISA is not a molecular test
Line 160 I think benefit fits here better
Line 171 enterovirus or rhinovirus species
Line 212-219: the authors only mention one tool for data analysis and in-house analysis only
Line 235 the methodology should be explained
Line 335 the reasoning why host nucleic acid depletion would only be suitable for a subset of samples and only for DNA is not clearly explained
Line 340-341 it is not clear what is meant by bric-a-brac DNA
Line 352 reference should be adjusted
Line 406 I don’t see the added value of discussing this separately.
Line 417-418 taxonomy has historically been based upon genetic information with often additional biological properties. A recent consensus statement recommends the classification upon sequencing data alone. An evolutionary relationship underlines most taxonomic classifications. While there is high genetic diversity within the genus Mammarenavirus, this is not because they are classified upon clinical picture rather than genetic distance.
Line 562 These are examples of factors explaining the high genetic diversity of viruses. Numerous of others mechanisms also factor in to this. It is not clear why these are singled out.
Line 502-518 The presented tools are rather limited. If the focus of this review lies in diagnostics, tools specific for human pathogens like Pathseq should be discussed.
Line 598-601 In a diagnostic setting, turn-around times are pivotal. The main focus should be that next to long reads, ONT is a platform for sample to diagnosis in real time.
Line 516: the possible use of de novo assembly has not been presented
Line 582-678 A strong future advantage in a diagnostic setting is automated library preparation using VolTRAX. I would suggest the authors discuss this as well.